# Stability of Buriti Oil Microencapsulated in Mixtures of Azuki and Lima Bean Flours with Maltodextrin

**DOI:** 10.3390/foods13131968

**Published:** 2024-06-21

**Authors:** Caroline Gregoli Fuzetti, Vânia Regina Nicoletti

**Affiliations:** Institute of Biosciences, Humanities and Exact Sciences (IBILCE), UNESP—São Paulo State University, Cristóvão Colombo Street, 2265, São José do Rio Preto 15054-000, SP, Brazil; vania.nicoletti@unesp.br

**Keywords:** pulses, microencapsulation, carotenoids, *Vigna angularis* L., *Phaseolus lunatus* L., *Mauritia flexuosa* L.

## Abstract

Buriti oil (*Mauritia flexuosa* L.) is rich in carotenoids, mainly β-carotene, and has great value for application as a food, pharmaceutical, or cosmetic ingredient, as well as a natural pigment. Microencapsulation is a promising technique to protect compounds sensitive to degradation such as β-carotene. Materials composed of carbohydrates and proteins, such as azuki bean (*Vigna angularis* L.) and lima bean (*Phaseolus lunatus* L.) flours, are alternative matrices for microencapsulation, which additionally provide good amounts of nutrients. In combination with maltodextrin, the flours represent a protective barrier in stabilizing lipophilic compounds such as buriti oil for subsequent spray drying. In this work, the performance of mixtures of maltodextrin with whole azuki and lima bean flours was evaluated in the microencapsulation of buriti oil. The microcapsules showed good results for solubility (>80%), hygroscopicity (~7%), encapsulation efficiency (43.52 to 51.94%), and carotenoid retention (64.13 to 77.49%.) After 77 days of storage, the microcapsules produced maintained 87.79% and 90.16% of carotenoids, indicating that the powders have high potential for application as encapsulants in the food and pharmaceutical industries.

## 1. Introduction

Carotenoids are natural pigments found in many foods and recognized for their bioactive properties such as antioxidant and anticarcinogenic activities, in addition to being precursors of vitamin A, which may offer health benefits associated with these compounds [1]. The oil extracted from the buriti palm fruit (*Mauritia flexuosa* L.) is of great interest for the food, chemical, and cosmetic industries due to the high concentration of carotenoids, with β-carotene being the major carotenoid in its composition [2]. In addition to being used as a dye, β-carotene has preventive and therapeutic health benefits, acting in the prevention of cardiovascular, degenerative, and immunological diseases [3].

The application of β-carotene may be limited as they are carotenoids susceptible to degradation when exposed to different conditions such as light, oxygen, and heat, leading to loss of benefits. In addition, hydrophobicity is a limitation that hinders its incorporation into water-based foods, due to low solubility [4]. Microencapsulation is a technology that allows the protection of sensitive bioactive compounds while helping to maintain their nutritional and sensory characteristics, thus improving the stability of the material, as well as assisting in the dispersion of poorly soluble materials [3,5].

Microencapsulation by atomization through spray drying is a widely used method to microencapsulate oils and hydrophobic compounds, such as carotenoids [2,6]. The active material is dispersed in a solution of carrier agents and the droplets are atomized through a nozzle in a drying chamber, where the water will be vaporized in contact with hot air, forming dry particles that trap the active compound within the matrix [7].

Mixtures of proteins and polysaccharides are often used to compose encapsulating matrices [6]. The combination of proteins and polysaccharides is advantageous because while proteins act as emulsifiers and form a protective layer, carbohydrates form the scaffold of the matrix [8].

Among polysaccharides, maltodextrin stands out due to its ease of application, low cost, solubility, neutrality, high availability, and heat resistance [9]. Maltodextrins provide good oxidative stability to the microencapsulated oil, acting as a barrier against oxygen. However, it is a material with low emulsifying capacity. In this case, maltodextrins are often associated with other encapsulating materials such as proteins and gums. The mechanisms by which maltodextrin affects the physicochemical properties of buriti oil microcapsules may include the formation of a polymeric matrix around the oil droplets and the ability of maltodextrin to adsorb and retain water, thus influencing the stability and structure of the microcapsules [6,10].

Within the group of proteins, whey and soy proteins are often combined with maltodextrin to form encapsulation matrices [11]. Emulsification of lipids in a carrier solution is an essential procedure for encapsulating lipophilic compounds, such as oils, carotenoids, and hydrophobic vitamins [12]. The electrostatic interaction between polysaccharides and proteins contributes to emulsion stabilization [13]. Proteins act as surfactants, facilitating the formation of the oil-in-water emulsion and increasing its stability [14]. During homogenization of oil in water, proteins reduce surface tension by adsorbing at the interface, creating a protective layer around the oil droplets [15]. Polysaccharides, on the other hand, interact with the proteins to increase the stabilizing layer thickness. Hydrophilic chain segments of polysaccharides may also protrude from the droplet surface, enhancing steric stabilization and binding great amounts of water, and thus contributing to emulsion stabilization by increasing viscosity [16]. Purified protein fractions, such as isolates and concentrates, have been investigated, expanding the opportunities for their use as encapsulation matrices. A promising combination for the microencapsulation of bioactive compounds is the use of maltodextrin blended with plant proteins extracted from legumes [2,17].

Vegetable proteins have low cost, lower allergenicity, and appropriate functional properties of emulsification for encapsulation purposes [18]. Bean grains provide important amounts of essential nutrients to humans, such as proteins (20–30%) and complex carbohydrates, including starch, dietary fiber, and oligosaccharides (55–75%), representing the main source of proteins for the populations of developing countries [19]. Azuki bean (*Vigna angularis* L.) and lima bean (*Phaseolus lunatus* L.) are foods that have been studied in recent years and present potential for exploration as sources of proteins and carbohydrates to be used as wall materials for microencapsulation [20,21].

Azuki bean grains are rich in polyphenols capable of acting as natural antioxidants. They are also a significant source of carbohydrates, proteins, and fiber, with a low lipid content. Studies show that azuki bean extract, rich in polyphenols, can help prevent obesity, inhibiting the accumulation of fat and improving metabolism. The health benefits of these grains are varied and important for public health issues [20,22,23]. Lima beans are a valuable source of nutrients for humans and animals, especially in developing countries. Its grains have a high nutritional value, with a good content of proteins, carbohydrates, and fiber, in addition to being rich in essential amino acids and minerals. This turns lima beans into an excellent source of plant protein with potential for application in foods [21,24].

Based on these considerations, this study proposes an innovative approach by combining whole azuki and lima bean flours with maltodextrin as matrices for microencapsulation of buriti oil. This combination offers a promising alternative for the formulation of encapsulating matrices, taking advantage of the functional properties of proteins and carbohydrates present in bean flours. In addition, we assessed the stability of the carotenoids microencapsulated through the spray drying process, studying the effect of time on the color and carotenoid content of the powders throughout the storage period.

## 2. Materials and Methods

### 2.1. Materials

The materials used in this work were as follows: azuki bean and lima bean grains purchased from local stores in the city of São Paulo (São Paulo, Brazil); maltodextrin DE 10 supplied by Get do Brasil (São João da Boa Vista, São Paulo, Brazil); buriti oil supplied by Amazon OilTM (Ananindeua, Pará, Brazil); and analytical grade reagents sodium hydroxide, hydrochloric acid, and chloroform supplied by Dinâmica (Indaiatuba, Brazil).

### 2.2. Methods

#### 2.2.1. Pre-Treatment of Grains and Flour Production

Azuki and lima bean grains were previously pre-treated via immersion in water (1 kg/5 L) at 25 °C for 12 h and then dried in an oven at 60 °C for 12 h. After pre-treatment, the beans were ground in a coffee grinder (J. R. Araújo & Cia Ltda, São Paulo, Brazil). The flours were sieved using a 35-mesh sieve to remove skins and impurities (Figure 1).

#### 2.2.2. Microencapsulation of Buriti Oil by Spray Drying

Emulsions of buriti oil with azuki and lima bean flours were formulated according to the proportions defined by Locali-Pereira et al. [17]. Two concentrations of flours were defined to prepare the emulsions, combined with maltodextrin as drying adjuvant (Table 1).

To prepare the emulsions, the flour was dispersed in water, adjusting the pH to 10 with 0.1 M NaOH to increase the protein solubility. Buriti oil was added and the mixture was homogenized (T-25 UltraTurraxR, IKA, Staufen, Baden-Württemberg, Germany) at 20,000 rpm for 10 min. Maltodextrin was added and the system was homogenized again for 10 min at 20,000 rpm. Finally, the emulsions were sonicated (Sonic Ruptor 4000, Omni International, Irmo, SC, USA) for 3 min (240 W/20 kHz) to reduce droplet size. The emulsions were dried in a mini spray dryer (B-290, Büchi, Meierseggstrasse, Flawil, Switzerland), with an atomizing nozzle of 0.7 mm in diameter, aspiration at 90%, feed flow at 6 mL/min, drying air flow at 819 L/h, and a drying air inlet temperature of 140 °C. After drying, the microencapsulated powders were stored in polyethylene bags with zip lock closure, protected from light with aluminum foil and stored in a desiccator containing silica gel.

#### 2.2.3. Characterization of Microcapsules

The buriti oil microcapsules were characterized through analysis of morphology, moisture, aW, hygroscopicity, solubility, process yield, colorimetric analysis, oil and carotenoid encapsulation efficiency, and oil and carotenoid retention.

##### Scanning Electron Microscopy (SEM)

The microcapsules were analyzed by scanning electron microscopy (SEM) (Philips XL-30 FEG, Hillsboro, OR, USA) at magnitudes of 500×, 5000×, and 10,000×. The samples were fixed on SEM stubs with double-sided adhesive and coated with gold/palladium under vacuum.

##### Moisture Content

The moisture content was determined using the gravimetric method in a vacuum oven at 70 °C and 100 mmHg (3.3 kPa) (AOAC, 926.12) (AOAC, 1990) [25]. Approximately 3 g of each sample were weighed into crucibles and placed in the oven for 48 h. Moisture was calculated on a dry basis (d.b.), according to Equation (1):(1)$$Moisture\left(\%\right)=\frac{W_{0}-W_{f}}{W_{f}}\times 100$$
in which W_0_ and W*_f_* are the initial and final weight (g) of the sample, respectively.

##### Water Activity

Water activity (aW) was measured using an electronic water activity meter (Aw Sprint, Novasina, Axair Ltd., Talstrasse, Pfäffikon, Switzerland), at 25 °C.

##### Hygroscopicity

Hygroscopicity was determined according to the methodology proposed by de Barros Fernandes et al. [26]. The samples (about 1 g) were weighed (W_0_) into small polyethylene containers and placed in a small chamber containing saturated NaCl solution (75.3% RH), at 25 °C, for 7 days. After this period, the samples were weighed again (W*_f_*). The hygroscopicity (%) was expressed as the percentage of moisture adsorbed, according to Equation (2):(2)$$Hygroscopicity\left(\%\right)=\frac{W_{f}-W_{0}}{W_{0}}\times 100$$

##### Solubility

The solubility was determined according to the methodology proposed by Cano-Chauca et al. [27]. Approximately 1 g of sample was dispersed in a beaker containing 100 mL of distilled water at 25 °C and stirring on a magnetic stirrer at high speed for 5 min. The dispersion was transferred to a Falcon tube and centrifuged at 3000× *g* for 5 min. A 25 mL-aliquot of the supernatant was collected and transferred to a previously weighed Petri dish, and dried in an oven at 105 °C for 5 h. The Petri dish was weighed again and the solubility (%) was calculated by the ratio between the weight of residue present in the dish (W*_f_*) and the sample weight (W_0_), according to Equation (3):(3)$$Solubility\left(\%\right)=\frac{W_{f}}{W_{0}}\times 100$$

##### Process Yield

The process yield was calculated according to Fuzetti et al. [28], by the ratio between the weight of powder obtained in the spray dryer collector after drying (W*_f_*) and the weight of total solids in the sample before drying (W_0_), according to Equation (4):(4)$$Process\;yield\left(\%\right)=\frac{W_{f}}{W_{0}}\times 100$$

##### Colorimetric Analysis

The colorimetric analysis of microcapsules was carried out using a CR-5 colorimeter (Konica Minolta, Ransey, NJ, USA), with observer at 10° and D65 illuminant. The determined color parameters were L* (Luminosity), C* (Chroma), h° (tone angle), a* (+a = red; −a = green), and b* (+b = yellow; −b = blue).

The color difference (∆E) between samples was determined using Equation (5) [29].
(5)$$∆E=\sqrt{{\left(L^{*}-\mathrm{Lo}\right)}^{2}+{\left(a^{*}-\mathrm{ao}\right)}^{2}+{\left(b^{*}-\mathrm{bo}\right)}^{2}}$$
where Lo, ao and bo are the color parameters of the control sample and $L^{*}$, $a^{*}$, and $b^{*}$ are the color parameters of the comparison sample.

To measure the color of microcapsules using the colorimeter, a circular glass cuvette, specific for powdered products, was used. Before measurements, the colorimeter was calibrated with zero calibration (calibration 0%) and white calibration (calibration 100%).

##### Encapsulation Efficiency (EEC) and Retention of Carotenoids (RTC)

The carotenoid encapsulation efficiency (EEC) and carotenoid retention (RTC) of the buriti oil present in the microcapsules were evaluated according to the methodology proposed by Ribeiro et al. [6], with some modifications. The initial content of carotenoids in buriti oil (CI), the carotenoids present on the surface of the microcapsule (CS), and the total carotenoids present in the microcapsules (CT) were quantified. CI was determined by weighing an aliquot of buriti oil in a 10 mL volumetric flask, completing the volume with chloroform. The absorbance of the sample was measured at 465 nm (Biospectro SP220) and the carotenoids were expressed as β-carotene, according to Equation (6).
(6)$$C\;(µg/g)=\frac{Abs\times V\times 10^{4}}{2396\times m}$$
where C is the concentration of carotenoids (µg β-carotene/g sample), Abs is the absorbance reading (A), V is the dilution volume (mL), and m is the sample weight (g).

To quantify CS, an aliquot of 0.1 g of microcapsules was weighed in a test tube and added to 10 mL of chloroform, stirring manually for 1 min. Subsequently, the organic phase was filtered through filter paper, the absorbance of the aqueous phase was measured at 465 nm, and the carotenoids were determined using Equation (2), being expressed as *β*-carotene. CT was determined by adding 0.1 g of sample to 5 mL of McIlvaine’s buffer solution (pH 7.4) in a test tube that was vortexed for 30 s. After 12 h at rest, the mixture was added with 5 mL of chloroform and vortexed again for 30 s. Separation of the aqueous and organic phases was carried out via centrifugation at 1800 rpm for 10 min. The absorbance of the aqueous phase was measured at 465 nm and carotenoids were determined using Equation (2), being expressed as *β*-carotene.

The carotenoid encapsulation efficiency (EEC) and carotenoid retention (RTC) were calculated using Equations (7) and (8), respectively.
(7)$$EEC\;(\%)=\frac{\left(CT-CS\right)}{CT}\times 100$$
(8)$$RTC\;(\%)=\frac{\left(CT\times m_{powder}\right)}{\left(CI\times m_{oil}\right)}\times 100$$
in which CI is the total carotenoids present in the oil, *m_powder_* is the mass of microcapsules, and *m_oil_* is the mass of oil added to the emulsion.

##### Oil Encapsulation Efficiency (EEO) and Oil Retention (RTO)

The oil encapsulation efficiency (EEO) and the oil retention (RTO) were evaluated by the methodology proposed by Ribeiro et al. [6] with some modifications. To quantify the oil present on the surface of the microcapsules (OS), 0.1 g of sample was weighed in a test tube and added with 10 mL of chloroform, stirring manually for 1 min. Subsequently, the organic phase of the mixture was filtered through filter paper, transferred to a Petri dish, and dried in an oven at 60 °C until constant weight. The residue was considered as the surface oil. The total oil (OT) was determined by weighing 0.1 g of sample and combining with 5 mL of McIlvaine’s buffer solution (pH 7.4) in a test tube, vortexing for 30 s. Subsequently, the mixture was kept at rest for 12 h, after which 5 mL of chloroform were added to the tube, vortexing again for 30 s. Separation of the aqueous and organic phases was carried out by centrifugation at 1800 rpm for 10 min. The organic phase was transferred to a Petri dish and dried in an oven at 60 °C until constant weight. The residue was considered as the total oil. The oil encapsulation efficiency (EEO) and oil retention (RTO) were calculated using Equations (9) and (10), respectively.
(9)$$EEO\;(\%)=\frac{\left(OT-OS\right)}{OT}\times 100$$
(10)$$RTO\;(\%)=\frac{\left(OT\times m_{powder}\right)}{OI}\times 100$$
in which OI is the oil present in the emulsion.

##### Stability of Microcapsules

Samples of microcapsules were weighed in petri dishes (about 5 g) and stored in a sealed desiccator containing a saturated MgCl_2_ solution (constant relative humidity of 33%), in a light-protected environment at 25 °C. This temperature was chosen to reproduce common commercial storage conditions. The color parameters, moisture, and total carotenoids content in the microcapsules were evaluated every 7 days, over 77 days.

##### Statistical Analysis

Statistical analyzes were performed using Minitab 17 software. The results were subjected to analysis of variance (ANOVA), applying Tukey’s multiple comparison test at a significance level of 5%.

## 3. Results and Discussion

### 3.1. Characterization of Microcapsules

#### 3.1.1. Scanning Electron Microscopy (SEM)

In general, azuki and lima bean flours combined with maltodextrin resulted in particles with a rounded shape, smooth surfaces with small wrinkles, and no porosities (Figure 2), which contributes to improving the stability of the encapsulated oil [17]. These aspects are characteristic of microcapsules produced by spray drying and may be associated with shrinkage resulting from the significant loss of moisture in the process [30].

This study revealed that formulations A1 and L1 were less prone to agglomeration compared to formulations A2 and L2, which were prone to agglomeration. The literature indicates that during microencapsulation by spray drying, it is possible for smaller particles to agglomerate around larger particles, which can provide greater stability to the microencapsulated compounds, as the external particles have the potential to protect the inner particles [31]. It is still possible to observe that lima bean flour (Figure 2c,d) resulted in larger microcapsules when compared to microcapsules produced with azuki bean flour (Figure 2a,b).

#### 3.1.2. Physicochemical Properties

In general, the combination of azuki bean and lima bean flours with maltodextrin enabled the formation of microcapsules rich in carotenoids and with good properties, which are relevant for their functionality (Table 2, Figure 3).

Regarding properties related to shelf life of microcapsules, such as moisture content, aW, and hygroscopicity, the results showed that the spray drying process was effective in producing powders with low humidity (less than 3%). The flours of both types of beans used in the formulations did not significantly influence the moisture content of the microcapsules (*p* > 0.05). The low values observed are typical in powder products resulting from spray drying due to the transformation of fluids into small droplets during fast drying in the chamber at high temperatures [32]. According to Ramakrishnan et al. [33], to assure the physicochemical stability, a powdered product should have a water content smaller than 5%.

As observed in Table 2, the formulations presented low values for aW (~0.2), which is desirable to avoid microbiological growth and to delay oxidation reactions in the product [34]. The powders presented hygroscopicity values of around 7%, close to the results obtained for buriti oil microcapsules with maltodextrin and kidney bean and moyashi bean flours [17]. Low hygroscopicity values, as low as those reported in Table 2, are recognized as capable of ensuring physicochemical and microbiological stability of powdered foods [35].

The microcapsules showed high solubility in water (above 80%), which is desirable for dissolution in aqueous food formulations. According to Jafari et al. [36], the solubility of microcapsules can be influenced by several factors, such as the raw material used, the carrier agents used, and the specific conditions during the microencapsulation process, especially temperature, air flow rate, and flow rate food. For both types of beans, a slightly higher solubility was observed in formulations with higher protein content (A2, L2). The protein fractions of grains such as cereals and legumes have good amphiphilic and functional properties, such as water solubility [37].

The drying yield was similar (no significant differences at *p* < 0.05) for the formulations with azuki and lima bean flours, ranging from 56 to 60%. These values are consistent with similar oil/water emulsions dried via spray drying at laboratory scale [6,17].

Regarding color parameters, as seen in Figure 3 and Table 2, the color of the microcapsules was influenced by the buriti oil and type of flour. Samples A1, A2, formulated with maltodextrin and azuki bean flour, showed significantly lower luminosity (L*), whereas samples L1, L2, with maltodextrin and lima bean flour, had higher luminosity (L*). In general, the microcapsules had a yellow tone (indicated by the hue angle h°~80 in the first quadrant) and a predominance of the positive b* coordinate tending towards yellow, based on the reference axis: 0° or 360° (red), 90° (yellow), 180° (green), and 270° (blue) [38].

The total oil retention (RTO) and carotenoid retention (RTC) are parameters related to the total content of oil and carotenoids present in the microcapsules compared to the formulated emulsion before drying [6]. On the other hand, the carotenoid (EEC) and oil (EEO) encapsulation efficiency are related to the content of carotenoids and oil present inside and outside the microcapsules. In the present study, RTO varied from 64.04 to 73.57%, with no significant difference between the samples (*p* < 0.05), whereas EEO varied from 60.73 to 68.67% and was higher for formulation L2, with lima bean flour at a higher protein content. Carotenoid retention (RTC) ranged from 64.13 to 77.49% while carotenoid encapsulation efficiency (EEC) ranged from 43.52 to 51.94%, both parameters being lower than RTO and EEO, respectively. These results may be explained by carotenoid losses that may have occurred during the drying process, as well as because the results were expressed in terms of β-carotene, and other carotenoids that may be present in buriti oil were not considered.

The results were notably influenced by the type and concentration of flour in the formulations. In microcapsules containing azuki bean flour, it was observed that the formulation with the lowest concentration of flour presented higher EEO, RTO, EEC, and RTC values. On the other hand, for microcapsules containing lima bean flour, the formulation with the highest flour content demonstrated the highest results for EEO, RTO, EEC, and RTC. Lima bean flour is characterized by a high amount of amylose in its composition, which can give it good gelling capacity [39]. The combination of a greater amount of flour probably provided a better surface barrier, reducing the loss of oil and carotenoids for the L2 formulation.

In this study, microcapsules of buriti oil stabilized with whole azuki and lima bean flours were obtained, presenting EEO, RTO, EEC, and RTC values like those found by other researchers who microencapsulated buriti oil in matrices combined with proteins and carbohydrates [6,17,40].

#### 3.1.3. Stability of Microcapsules

Considering the results obtained for physicochemical properties, both flours allowed the production of microcapsules rich in carotenoids and with adequate properties. Taking into account a hypothetical greater protection that could result from the morphological characteristics of the powders and the higher protein concentration, formulations A2 (with 10% azuki bean flour) and L2 (with 10% lima bean flour) were selected to be evaluated regarding their storage stability, which was followed for 77 days at 25 °C and 33% relative humidity (Figure 4).

As seen in Figure 4, there were significant reductions (*p* < 0.05) in the total carotenoid content present in the microcapsules along the 77-day storage period, although the total carotenoid content remained almost stable in the first 7 days of storage. Nevertheless, the total carotenoid reductions can be considered relatively small: for microcapsules with azuki bean flour, there was a 12.21% reduction at the end of the 77th day, while microcapsules formulated with lima bean flour recorded a 9.84% decrease in total carotenoid content during the same period. Vieira et al. [41] studied the stability of red pitaya (*Hylocereus polyrhizus*) pigment microencapsulated in maltodextrin matrices. After 90 days of storage, the authors observed a 14.28% reduction in total betalains content. Guadarrama-Lezama et al. [42] investigated the stability of carotenoids contained in chili extract (*Capsicum annuum* L.) microencapsulated in gum arabic and maltodextrin matrices and obtained maximum stability at 25 °C, in an aw range of 0.2 to 0.3, with better retention and preservation of total carotenoid content.

The reduction in carotenoid content during storage may have been caused by the increase in moisture content undergone by the samples, since moisture can act as a plasticizing agent, thus promoting physicochemical changes in the dry materials, including oxidation reactions. The moisture of the microcapsules increased along the storage period, ranging from 2.36 to 2.57% for samples with azuki bean flour and from 2.42 to 2.6% for samples with lima bean flour (Figure 4). However, the values remained below the 5% recommended to guarantee their physicochemical stability [33]. Although the moisture gain seems to be small, it can be enough to change the physical state of the wall materials, thus affecting the mobility of reactive species in the matrix. In addition, other factors, such as exposure to oxygen, chemical structure of the carotenoids, materials, and encapsulation conditions can influence significant differences [6,43,44].

The color analysis is essential to indicate the uniformity and the quality of a product. The values of the color parameters L*, a*, and b* obtained through the colorimeter were used to quantify the total color difference (ΔE) of the microcapsules during the storage period, compared to the initial samples (0 days of storage) (Table 3).

The color parameters remained almost unchanged from 0 to 7 days. Initially, the samples had a darker and more saturated color, but lost color intensity over time. In addition, there was an increase in the values of the luminosity coordinate (L*), whereas the a* and b* coordinates decreased significantly (*p* < 0.05), also resulting in a decrease in saturation, as indicated by the values of chroma parameter (C*).

According to Óbon et al. [45], total color difference (ΔE) values between 0 and 1.5 are considered low, and the difference between samples cannot be perceived visually. On the other hand, values between 1.5 to 5 indicate that the color difference can be noticed visually. Therefore, the total color variation (ΔE) between samples increased, indicating that the microcapsules became lighter due to the loss of carotenoids. These results are directly associated with the total carotenoid levels evaluated during the storage period.

Although there was statistical significance in the carotenoid content decrease, as well as changes in color parameters detected along the storage period, these reductions were modest and the microcapsules still retained a considerable amount of β-carotene, suggesting a great potential for their application as a natural pigment or functional ingredient in food products.

## 4. Conclusions

Buriti oil microcapsules with a high carotenoid content were produced using mixtures of maltodextrin with azuki and lima bean flours, obtaining morphological aspects typical of microcapsules produced by spray drying, characteristics that contribute to improving the stability of the encapsulated oil. However, lima bean flour resulted in larger microcapsules compared to azuki bean flour. Good results in terms of encapsulation efficiency, process yield, solubility, and hygroscopicity were obtained for the powders. The color properties of the microcapsules were influenced by the buriti oil and the type of flour used. The formulations with higher flour contents showed better characteristics and were selected for storage stability studies for 77 days. At the end of the period, a modest decrease in color parameters and carotenoid levels in microencapsulated buriti oil was observed. The results of this study highlight the viability of using azuki and lima bean flours in combination with maltodextrin to produce carotenoid-rich microcapsules and suggest considerable potential for the use of buriti oil microcapsules as a pigment or natural ingredient in the food and pharmaceutical industries.

## Figures and Tables

**Figure 1 foods-13-01968-f001:**
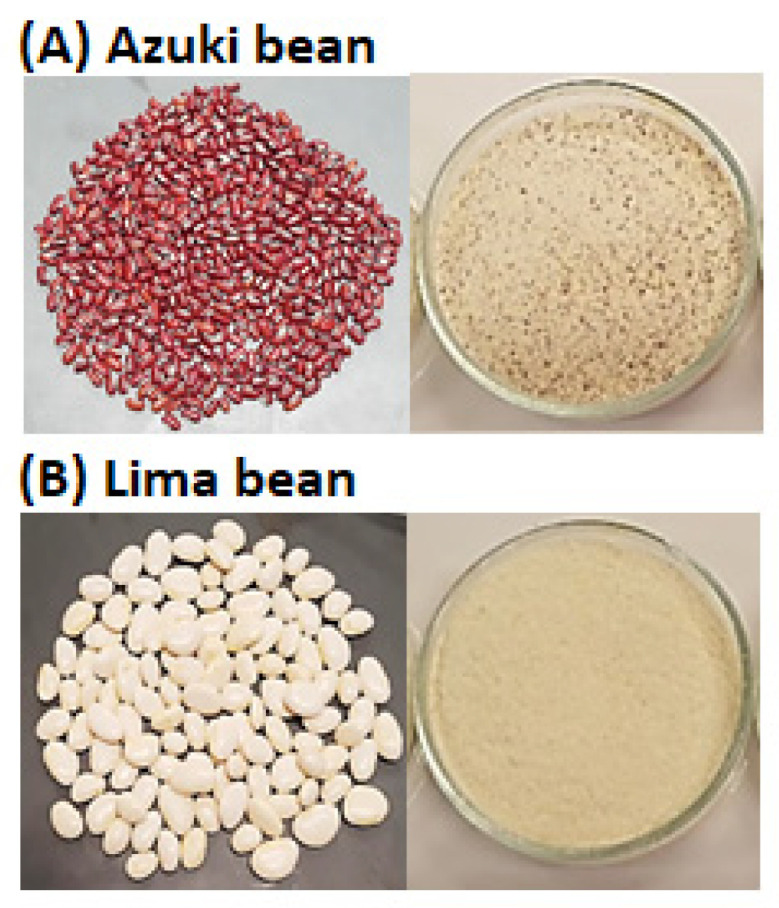
Grains and flours of azuki bean (**A**) and lima bean (**B**).

**Figure 2 foods-13-01968-f002:**
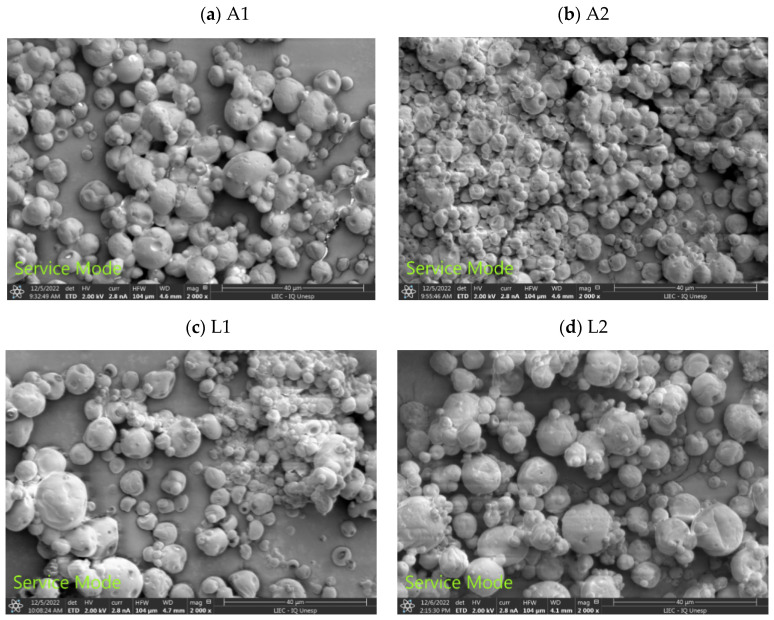
Scanning electron microscopy images (2000× magnification) of buriti oil microcapsules. A1, A2: microcapsules formulated with azuki bean flour equivalent to 5% and 10% in the formulation, respectively; L1, L2: microcapsules formulated with lima bean flour equivalent to 5% and 10% in the formulation, respectively.

**Figure 3 foods-13-01968-f003:**
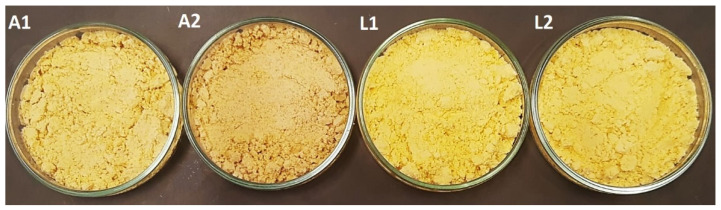
(A1, A2): microcapsules formulated with azuki bean flour equivalent to 5% and 10% in the formulation, respectively. (L1, L2): microcapsules formulated with lima bean flour equivalent to 5% and 10% in the formulation, respectively.

**Figure 4 foods-13-01968-f004:**
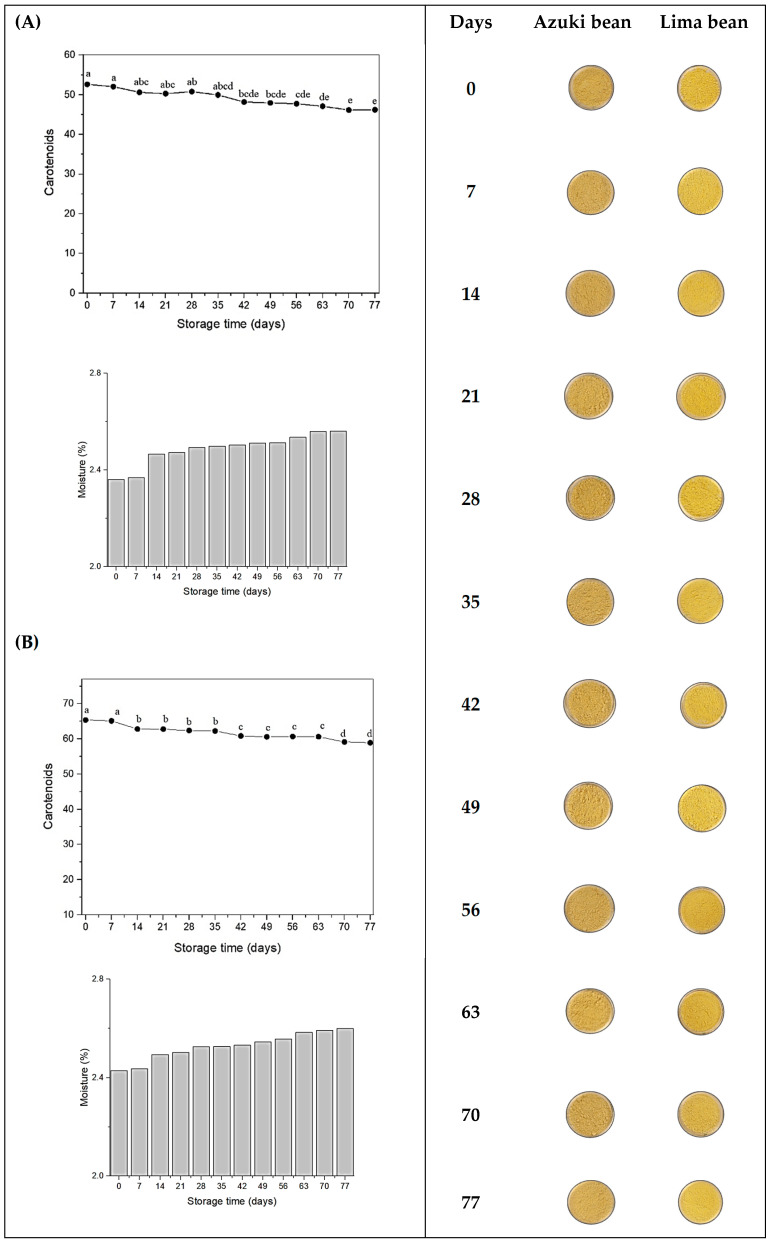
Total carotenoids content (µg of β-carotene/g sample), moisture (%), and images of microcapsules produced with azuki bean flour (**A**) and lima bean flour (**B**), stored for 77 days at 25 °C and 33% relative humidity. Different letters indicate statistical difference between days (*p* < 0.05).

**Table 1 foods-13-01968-t001:** Emulsion formulations with buriti oil, combined with maltodextrin, azuki bean (A), and lima bean (B) flours.

	Flour	Water	Buriti Oil	Maltodextrin
	(g/100 g of Emulsion)			
(A) Azuki bean	5	55	10	30
	10	55	10	25
(B) Lima bean	5	55	10	30
	10	55	10	25

**Table 2 foods-13-01968-t002:** Physicochemical properties of buriti oil microcapsules. A1, A2: microcapsules formulated with azuki bean flour equivalent to 5% and 10% in the formulation, respectively; L1, L2: microcapsules formulated with lima bean flour equivalent to 5% and 10% in the formulation, respectively. Means ± standard deviation followed by different letters in the lines indicate statistical difference (*p* < 0.05).

Properties	A1	A2	L1	L2
Moisture (%)	2.61 ± 0.21 ^A^	2.44 ± 0.04 ^A^	2.70 ± 0.06 ^A^	2.38 ± 0.24 ^A^
aW	0.25 ± 0.09 ^AB^	0.23 ± 0.05 ^B^	0.26 ± 0.07 ^A^	0.24 ± 0.01 ^AB^
Hygroscopicity (%)	7.47 ± 0.13 ^B^	7.94 ± 0.17 ^A^	7.74 ± 0.01 ^AB^	7.78 ± 0.20 ^AB^
Solubility (%)	81.58 ± 0.26 ^B^	83.97 ± 0.14 ^A^	81.28 ± 0.18 ^B^	83.45 ± 1.09 ^A^
Process Yield (%)	60.97 ± 5.21 ^A^	57.9 ± 7.5 ^A^	58.28 ± 4.04 ^A^	56.50 ± 3.54 ^A^
L*	84.36 ± 0.32 ^B^	80.77 ± 0.58 ^C^	86.88 ± 0.44 ^A^	86.45 ± 0.72 ^A^
a*	3.58 ± 0.16 ^A^	3.56 ± 0.25 ^A^	3.46 ± 0.26 ^A^	3.10 ± 0.10 ^B^
b*	30.78 ± 0.72 ^A^	27.09 ± 1.14 ^B^	31.67 ± 1.36 ^A^	30.70 ± 1.03 ^A^
C*	30.99 ± 0.73 ^A^	27.32 ± 1.16 ^B^	31.86 ± 1.38 ^A^	30.85 ± 1.02 ^A^
h°	83.36 ± 0.16 ^C^	82.50 ± 0.23 ^D^	83.75 ± 0.23 ^B^	84.21 ± 0.30 ^A^
EEO (%)	61.91 ± 0.20 ^B^	60.73 ± 0.62 ^B^	62.21 ± 3.12 ^AB^	68.67 ± 0.62 ^A^
RTO (%)	68.91 ± 1.92 ^A^	64.04 ± 0.87 ^A^	64.82 ± 0.96 ^A^	73.57 ± 4.73 ^A^
EEC (%)	51.31 ± 3.87 ^AB^	43.52 ± 3.33 ^C^	44.09 ± 1.64 ^BC^	51.94 ± 1.55 ^A^
RTC (%)	76.76 ± 6.97 ^A^	64.13 ± 4.80 ^B^	65.64 ± 2.19 ^AB^	77.49 ± 3.22 ^A^

**Table 3 foods-13-01968-t003:** Color parameters for buriti oil microcapsules formulated with azuki bean flour (**A**) and lima bean flour (**B**), stored for 77 days at 25 °C and 33% relative humidity. Means ± standard deviation followed by different letters in the columns indicate statistical difference (*p* < 0.05).

**(A)**						
**Days**	**L***	**a***	**b***	**C**	**h°**	**ΔE**
0	83.54 ± 0.35 ^e^	3.30 ± 0.15 ^a^	25.97 ± 0.54 ^a^	26.18 ± 0.55 ^a^	82.74 ± 0.18 ^e^	
7	83.53 ± 0.01 ^e^	3.34 ± 0.015 ^a^	26.18 ± 0.02 ^a^	26.39 ± 0.02 ^a^	82.72 ± 0.03 ^e^	0.20 ± 0.02 ^f^
14	84.20 ± 0.09 ^cd^	2.89 ± 0.032 ^bc^	24.78 ± 0.13 ^b^	24.96 ± 0.13 ^b^	83.33 ± 0.05 ^d^	1.42 ± 0.16 ^e^
21	84.36 ± 0.04 ^abc^	2.76 ± 0.02 ^cd^	24.26 ± 0.03 ^c^	24.42 ± 0.03 ^c^	83.50 ± 0.04 ^cd^	1.97 ± 0.05 ^d^
28	84.23 ± 0.1 ^bcd^	2.89 ± 0.41 ^bc^	24.92 ± 0.08 ^b^	25.09 ± 0.09 ^b^	83.37 ± 0.07 ^cd^	1.32 ± 0.14 ^e^
35	83.98 ± 0.01 ^d^	2.92 ± 0.011 ^b^	24.95 ± 0.01 ^b^	25.12 ± 0.01 ^b^	83.31 ± 0.02 ^d^	1.17 ± 0.01 ^e^
42	84.68 ± 0.05 ^a^	2.72 ± 0.02 ^de^	24.15 ± 0.13 ^cd^	24.30 ± 0.13 ^cd^	83.55 ± 0.03 ^c^	2.23 ± 0.14 ^cd^
49	84.51 ± 0.02 ^abc^	2.55 ± 0.02 ^fg^	23.91 ± 0.07 ^cd^	24.05 ± 0.07 ^cd^	83.91 ± 0.03 ^ab^	2.40 ± 0.07 ^bc^
56	84.54 ± 0.01 ^ab^	2.60 ± 0.011 ^ef^	24.16 ± 0.02 ^cd^	24.30 ± 0.02 ^cd^	83.83 ± 0.02 ^ab^	2.18 ± 0.02 ^cd^
63	84.52 ± 0.03 ^abc^	2.70 ± 0.03 ^de^	23.67 ± 0.10 ^de^	23.83 ± 0.09 ^de^	83.47 ± 0.04 ^cd^	2.57 ± 0.10 ^b^
70	84.43 ± 0.02 ^abc^	2.51 ± 0.03 ^fg^	23.25 ± 0.08 ^e^	23.38 ± 0.08 ^e^	83.81 ± 0.05 ^b^	2.97 ± 0.08 ^a^
77	84.70 ± 0.05 ^a^	2.44 ± 0.03 ^g^	23.33 ± 0.09 ^e^	23.45 ± 0.10 ^e^	84.03 ± 0.06 ^a^	3.01 ± 0.11 ^a^
**(B)**						
**Days**	**L***	**a***	**b***	**C**	**h°**	**ΔE**
0	88.19 ± 0.03 ^e^	2.39 ± 0.03 ^b^	27.79 ± 0.02 ^b^	27.90 ± 0.03 ^b^	85.08 ± 0.07 ^f^	
7	87.61 ± 0.05 ^f^	2.56 ± 0.08 ^a^	28.25 ± 0.28 ^a^	28.37 ± 0.29 ^a^	84.81 ± 0.10 ^g^	0.77 ± 0.23 ^e^
14	88.63 ± 0.06 ^a^	2.17 ± 0.02 ^c^	26.63 ± 0.07 ^c^	26.72 ± 0.07 ^c^	85.32 ± 0.04 ^e^	1.26 ± 0.08 ^d^
21	88.38 ± 0.03 ^d^	2.04 ± 0.02 ^d^	26.51 ± 0.08 ^c^	26.59 ± 0.08 ^c^	85.59 ± 0.04 ^d^	1.34 ± 0.09 ^d^
28	88.36 ± 0.03 ^d^	2.03 ± 0.05 ^d^	26.33 ± 0.02 ^c^	26.41 ± 0.02 ^c^	85.57 ± 0.01 ^d^	1.51 ± 0.02 ^d^
35	88.33 ± 0.02 ^d^	1.82 ± 0.02 ^e^	25.72 ± 0.12 ^d^	25.78 ± 0.12 ^d^	85.94 ± 0.05 ^c^	2.15 ± 0.11 ^c^
42	88.40 ± 0.02 ^cd^	1.97 ± 0.07 ^de^	25.61 ± 0.13 ^de^	25.69 ± 0.14 ^de^	85.58 ± 0.14 ^d^	2.22 ± 0.14 ^c^
49	88.34 ± 0.04 ^d^	1.67 ± 0.02 ^fg^	25.34 ± 0.04 ^ef^	25.40 ± 0.04 ^ef^	86.22 ± 0.05 ^ab^	2.55 ± 0.04 ^b^
56	88.36 ± 0.01 ^d^	1.76 ± 0.01 ^ef^	25.30 ± 0.06 ^ef^	25.37 ± 0.06 ^ef^	86.01 ± 0.02 ^bc^	2.57 ± 0.05 ^b^
63	88.49 ± 0.02 ^bc^	1.63 ± 0.02 ^g^	25.24 ± 0.04 ^f^	25.30 ± 0.04 ^f^	86.30 ± 0.05 ^a^	2.67 ± 0.03 ^b^
70	88.40 ± 0.01 ^cd^	1.58 ± 0.04 ^g^	24.71 ± 0.08 ^g^	24.76 ± 0.08 ^g^	86.33 ± 0.08 ^a^	3.19 ± 0.08 ^a^
77	88.51 ± 0.01 ^b^	1.57 ± 0.04 ^g^	24.55 ± 0.07 ^g^	24.60 ± 0.07 ^g^	86.33 ± 0.09 ^a^	3.36 ± 0.07 ^a^

## Data Availability

The original contributions presented in the study are included in the article, further inquiries can be directed to the corresponding author.

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
