# Peer review of "Stability of Buriti Oil Microencapsulated in Mixtures of Azuki and Lima Bean Flours with Maltodextrin"

_foods, 2024, doi:10.3390/foods13131968_

Round 1
Reviewer 1 Report
Comments and Suggestions for Authors
This paper titled “Stability of Buriti Oil Microencapsulated in Mixtures of Azuki and Lima Bean Flours with Maltodextrin” fabricated the buriti oil loaded microcapsules combined whole meal flours obtained from azuki and lima beans with maltodextrin as matrices, and to evaluate the characteristics, encapsulation efficiency and stability of the obtained microcapsules, which performed great potentials for application as encapsulants in the food industry. However, there are some issues that should still be improved, and this manuscript can be considered for major revisions.
Other comments and advices are listed below.
1. How does the maltodextrin affect the physicochemical properties of the obtained buriti oil microcapsules and what are the mechanisms for these?
2. The flowability and bulk density of the obtained buriti oil microcapsules should be determined and compared.
3. What are the influences of the azuki and lima bean flours combined with maltodextrin on the powder size and size distribution of the buriti oil microcapsules?
4. What are the oxidation stability and other function characteristics (such as digestibility and bioavailability) of the microencapsulated buriti oil and carotenoids?
5. In Figure 3, the charts and images are not clear, which should be improved.
6. The “Materials and Methods” section should be in front of the “Results and Discussion”.
7. In Materials and Methods, the detailed characterization methods should be presented, such as Water Activity, Process Yield, etc.
8. The authors should check the manuscript carefully for spelling and grammatical errors. The sentence tense for the manuscript needed to be corrected carefully.
Comments on the Quality of English Language
Moderate. The English Language of the manuscript should be improved.
Author Response
Please see the attachment : Response to Reviewer 1 Comments

Reviewer 2 Report
Comments and Suggestions for Authors
The objective of this study was to combine whole meal flours obtained from azuki and lima beans with maltodextrin as matrices for the microencapsulation of buriti oil and to evaluate the characteristics of the microcapsules. In addition, assessed the stability of the carotenoids microencapsulated by the spray drying process, studying the effect of time on the color and carotenoid content of the powders throughout the storage period.
In general, the question is whether the order of the chapters is good: 1. Introduction, 2. Results and Discussion, 3. Materials and Methods, or 2. and 3. should switch places.
Line 42: need [2, 6]., instead of [6, 2].
Line 45: after [7]., you need a space.
What is novelty in this research?, should be better highlighted in the objectives.
The authors should explain why they chose the microcapsule test period to be a total of 77 days, every 7 days. On what basis were the investigations ended? Have there been significant changes in the investigated parameters? Which ones?
When colorimetric measuring the color of microcapsules, it is necessary to indicate which consumables was used to measure the color (eg cuvettes for powdery materials or others?).
It should also be stated how and with what the colorimeter was calibrated before the measurement (eg white standard plate).
Figure 3: Figure 3A and Figure 3B it would be better if they were shown in the picture together for azuki bean flour (A) and lima bean flour (B). In this way, the similarities and differences of microcapsules with azuki bean flour (A) and lima bean flour (B) would be better seen.
In Table 1, you should write above A1 and A2 azuki bean flour and and above L1 and L2 lima bean flour.
Tables 2 and 3 should be rearranged in the same way as Figure 3, so that the results for azuki bean flour (A) and lima bean flour (B).
It was not clearly stated which microcapsules were used to determine microcapsule stability, A1 or A2, L1 or L2, A1 and A2, combined, L1 and L2, combined, or something else.
The conclusion should be supplemented in accordance with the changes in Figure 3, and Tables 1, 2 and 3.
If possible, highlight the similarities/differences of microcapsules with azuki bean flour, i.e. lima bean flour.
Author Response
Please see the attachment : Response to Reviewer 2 Comments

Round 2
Reviewer 1 Report
Comments and Suggestions for Authors
The manuscript have been greatly improved and several problems have been well corrected. Overall, it can be accepted for the publication.
Reviewer 2 Report
Comments and Suggestions for Authors
The reviewers' comments were considered, and the authors made corrections to the manuscript in accordance with the comments.
I believe that the manuscript can be submitted to the further procedure for publication.